# Growth Factor Screening in Dystrophic Muscles Reveals PDGFB/PDGFRB-Mediated Migration of Interstitial Stem Cells

**DOI:** 10.3390/ijms20051118

**Published:** 2019-03-05

**Authors:** Jordi Camps, Hanne Grosemans, Rik Gijsbers, Christa Maes, Maurilio Sampaolesi

**Affiliations:** 1Laboratory of Translational Cardiomyology, Department of Development and Regeneration, Stem Cell Research Institute, KU Leuven, 3000 Leuven, Belgium; jordi.camps@kuleuven.be (J.C.); hanne.grosemans@kuleuven.be (H.G.); 2Department of Pharmaceutical and Pharmacological Sciences, Laboratory for Viral Vector Technology and Gene Therapy, KU Leuven, 3000 Leuven, Belgium; rik.gijsbers@kuleuven.be; 3Laboratory of Skeletal Cell Biology and Physiology (SCEBP), Skeletal Biology and Engineering Research Center (SBE), KU Leuven, 3000 Leuven, Belgium; christa.maes@kuleuven.be; 4Human Anatomy Unit, Department of Public Health, Experimental and Forensic Medicine, University of Pavia, 27100 Pavia, Italy

**Keywords:** chemokines, cell migration, PDGF-B, skeletal muscle, limb-girdle muscular dystrophy, stem cells

## Abstract

Progressive muscle degeneration followed by dilated cardiomyopathy is a hallmark of muscular dystrophy. Stem cell therapy is suggested to replace diseased myofibers by healthy myofibers, although so far, we are faced by low efficiencies of migration and engraftment of stem cells. Chemokines are signalling proteins guiding cell migration and have been shown to tightly regulate muscle tissue repair. We sought to determine which chemokines are expressed in dystrophic muscles undergoing tissue remodelling. Therefore, we analysed the expression of chemokines and chemokine receptors in skeletal and cardiac muscles from Sarcoglycan-α null, Sarcoglycan-β null and immunodeficient Sgcβ-null mice. We found that several chemokines are dysregulated in dystrophic muscles. We further show that one of these, platelet-derived growth factor-B, promotes interstitial stem cell migration. This finding provides perspective to an approachable mechanism for improving stem cell homing towards dystrophic muscles.

## 1. Introduction

Muscular dystrophies (MDs) are characterized by a chronic and progressive degeneration of both cardiac and skeletal muscles [1]. Because of the importance of stem cells in renewing the skeletal muscle, cell therapy holds an interesting potential for the treatment of MDs. While satellite cells (SCs) are the most potent skeletal muscle stem cell population, they do not possess any capacity for migration and extravasation. Mesoangioblasts (MABs), a subset of pericytes originating from the dorsal aorta, retain their potential of responding towards inflammatory cues like TNFα and HMGB1 and are able to efficiently migrate towards them [2,3]. Cell therapy with mesoangioblasts has shown a profound amelioration of skeletal muscle function in preclinical dystrophic animal models [4,5] and a first clinical trial confirmed the safety of MAB delivery procedures [6]. In the last decade, several groups have highlighted reciprocal functional interactions between different types of interstitial muscle cells in orchestrating muscle repair of injured muscles. In particular, signals released from fibro/adipogenic progenitors (FAPs) appear to be crucial to drive SCs towards myogenic differentiation [7].

Amelioration of skeletal muscle function, however, increases cardiac muscle burden in dystrophic conditions, thereby increasing the risk of heart failure [8]. In contrast to skeletal muscles, the heart is completely unable to regenerate and hence, undergoes an even worse degeneration. Sarcoglycan(sgc)-deficient mice genetically model limb-girdle MDs, yet they present many pathological characteristics that are also featured by Duchenne MD (DMD) patients [9]. Limb girdle muscular dystrophy (LGMD) patients often face cardiac complications, although this largely depends on the type [1]. LGMD type 2D patients lack Sarcoglycan-α and suffer from mild cardiomyopathy while patients with LGMD type 2E lack Sarcoglycan-β and suffer from severe cardiomyopathy [8]. Our group recently generated immunodeficient dystrophic mice (sgcb^−/−^; Rag2^−/−^; γc^−/−^; ID *Sgcb*-null mice) [10], which show a severe cardiomyopathy. Hence, it has now become possible to test the regenerative capacity of novel therapeutic approaches for MD on both muscle types concurrently.

Cardiac interstitial cells, such as cardiac mesoangioblasts [11], are an appealing cell therapy source because of their beneficial paracrine effects on cardiac remodelling [12]. However, when injected directly into the myocardium or skeletal muscles, stem cells show very limited migration potential. This limitation highlights the need of alternative routes of administration to aspire a greater overall engraftment. Unfortunately, even systemic delivery showed constraints due to the entrapment of delivered cells in the filter organs. In an effort to improve the therapeutic potential of cell therapy, in this study, we will focus on chemokines and growth factors released by dystrophic muscle and the potential to use them for a more targeted stem cell migration.

Notably, inflammation in the cardiac and skeletal muscle is a hallmark of MD, and several chemokines and growth factors were reported to be more abundant in the dystrophic muscles compared with the muscle of wild-type mice [13]. These chemokines have been shown to induce pericyte migration in the brain [14,15], the retina [16] and the heart [17] in response to acute and chronic tissue damage. Although several studies indicate that some environmental stimuli such as hypoxia and growth factors (e.g., SDF-1 and HMGB1) [2] influence cell migration, the signals and mechanisms governing pericyte migration are not yet fully understood.

In this perspective, we analysed the expression of chemotactic growth factors in skeletal and cardiac muscles of Sarcoglycan-α null, Sarcoglycan-β null and immunodeficient Sgcβ-null mice. We identified PDGF-B as a trigger for FAP and MAB cell migration. PDGF-B belongs to a family of four related growth factors involved in many biological processes. PDGF-B binds its receptor PDGFRB to activate downstream signalling pathways, controlling pericyte survival, migration, apoptosis, proliferation and differentiation [18]. Perturbing PDGFB/PDGFRB signalling using interstitial cells isolated from transgenic *Pdgfrb*-flox mice led to an impairment of migration. These results clarify the migration mechanism of interstitial cells and provide a novel therapeutic target for improving the efficacy of cell-based treatments.

## 2. Results

In order to find migration axes for interstitial stem cells, we performed a growth factor screening on healthy and dystrophic skeletal muscle. We performed qPCR on isolated RNA from the tibialis anterior muscle of C57Bl6 (Healthy) and *Sgca*-null (Dystrophic) mice (methods). We selected mice at the age of five months because from then, they start to describe signs of small foci of myocyte necrosis surrounded by atrophic and hypertrophic fibres, central nucleation and fibre splitting [19]. Primers were selected for 21 genes that are upregulated in mdx mice and were tested for efficiency (Appendix A) [13]. We found a general increase of the expression of chemokines and growth factors in dystrophic muscle (Figure 1A). We calculated the fold change difference in expression level per gene in dystrophic muscle compared to healthy muscle, ordered them in a descending manner and performed an independent *t*-test per gene between healthy and dystrophic samples (Figure 1B). Interestingly, we did not observe a downregulation of any tested gene, whereas several genes were significantly upregulated in dystrophic muscles. The highest fold change increases were observed for the CC-chemokine family such as *Ccl3, Ccl4, Ccl12, Ccl7, Ccl8, Ccl17, Ccl22, Ccl24, Ccl6, Ccl9* (Figure 1C). *Cx3cl1* or *Fractalkine* (*Fkn*) of the CX3C-family was also drastically increased in expression as well as *Sdf1* or *Cxcl12* of the CXC-family (Figure 1D). Finally, we also observed a rise in the expression of *Pdgfa* and *Pdgfb* in dystrophic muscle (Figure 1E). Altogether, we implemented a qPCR screening for growth factors involved in migration and found that a substantial amount of chemokines and growth factors were upregulated in the skeletal muscle of *Sgca*-null mice.

Due to the distinct levels of cardiomyopathy between patients lacking either Sarcoglycan-α or Sarcoglycan-β, we hypothesized that the growth factor expression profile may vary between these types. Therefore, we wanted to establish a growth factor footprint of dystrophic hearts from mice with varying cardiomyopathies. We isolated the heart from C57Bl6 (Wild-type), *Sgca*-null, immunocompetent (IC) *Sgcb*-null and immunodeficient (ID) *Sgcb*-null mice [10] and performed an RT^2^ profiler PCR array of 90 different chemokines and receptors (methods). We selected the mice at an age of seven months-old because from then, large areas of fibrosis are detectable in the heart [9]. All genes were normalized to four housekeeping genes (*Actb*, *Gapdh*, *Gusb* and *Hsp90ab1*). When we compared the chemokine expression of *Sgca*-null with Wild-type hearts, we found an overexpression of many CC-chemokine ligand and receptors like *Ccl7*, *Ccl8* and *Ccr2* and downregulation of *Cx3cr1* and *Cxcl10* (Figure 2A and Appendix A). Again, the CC-family chemokines and receptors were mainly found when we compared IC *Sgcb*-null with Wild-type hearts, however also inflammatory genes such as *Il6* and *Il1b* were observed. (Figure 2B). In contrast to Wild-type, the heart of ID *Sgcb*-null mice contained upregulated chemokines such as *Cx3cl1* and *Ccl8* as well as downregulated chemokines such as *Xcl1* (Figure 2C). The most differentially expressed (DE) genes were found in *Sgca*-null and IC *Sgcb*-null mice, while it concerned mainly upregulated genes. ID *Sgcb*-null mice contained both a low amount of down- and up-regulated genes (Figure 2D). When we compare DE genes in all dystrophic mice, we find that only a small fraction of genes are shared (Figure 2E). Further, to get an idea which chemokines are linked to cardiomyopathy and not the genetic deletion of the Sarcoglycan complex, we also checked the DE chemokines in IC *Sgcb*-null compared to *Sgca*-null mice. We detected mainly overexpressed genes such as *Cx3cl1* and can define these chemokines as specific for IC *Sgcb*-null mice (Figure 2F). Additionally, comparing the hearts of ID with IC *Sgcb*-null mice gives an idea about the chemokines that are expressed by immune cells. We only find downregulated chemokines and receptors (Figure 2G). When we clustered all significant DE genes, we observed five main groups; group 1 are genes that are not expressed in *Sgca*-null hearts, group 2 are genes that are upregulated in all forms of dystrophy and are immune system independent, group 3 genes are solely upregulated in the heart of *Sgcb*-null mice, group 4 consists of genes specific to the heart of *Sgca*-null mice and group 5 encompass genes that are not expressed in the heart of ID *Sgcb*-null mice (Figure 2H). We used gene set enrichment analysis with correlations to Gene Ontology (GO) for biological processes [20,21] on differentially up- and down-regulated genes between mice (Appendix A). We found different enriched GO terms for upregulated genes in *Sgca*-null and *Sgcb*-null mice with many terms linked to chemotaxis in *Sgcb*-null mice (Appendix A). Also, terms considering nitric oxide synthesis were connected to *Sgcb*-null mice, however not *Sgca*-null mice (Appendix A). GO terms concerning downregulated genes were quite similar between mice with the highest scores for ID *Sgcb*-null mice (Appendix A). In addition, we also checked the expression of the *Pdgf*-family genes in dystrophic compared to healthy hearts. We found a significant upregulation of *Pdgfb*, *Pdgfra* and *Pdgfrb* in IC and ID *Sgcb*-null heart compared to *Sgca*-null heart (Figure 2I). In conclusion, we performed a comprehensive chemokine expression array of different types of dystrophic hearts and relate specific clusters of chemokines to each genotype.

Now that we have a directory of chemokines and growth factors that are upregulated in dystrophic cardiac and skeletal muscles, we wanted to check which of the corresponding receptors is located on mesoangioblasts and fibro/adipogenic progenitors. Therefore, we isolated these cells from the hindlimbs of one to two months-old wild-type mice with fluorescent activated cell sorting (FACS) (methods). Mesoangioblasts were selected as Lin–Itga7–Alpl+Ly6a+ cells and fibro/adipogenic progenitors as Lin–Itga7–Alpl–Ly6a+ (Figure 3A). The gating’s were carefully selected with negative controls and a FMO (fluorescent minus one) for Alpl (Figure 3B and methods). We additionally characterized our cells further by qPCR and show that mesoangioblasts and fibro/adipogenic progenitors do not have any differentially expressed genes except for *Desmin* (Figure 3C). Furthermore, by flow cytometry, we demonstrated that a fraction of both mesoangioblasts and fibro/adipogenic progenitors are positive for CD34 and all are positive for CD44 (Figure 3D). In addition, we validated the differentiation potency of these cells by subjecting them to an adipogenic differentiation and a fusion co-culture assay with satellite cells (methods). Both mesoangioblasts and fibro/adipogenic progenitors were able to differentiate to adipocytes in vitro as well as fuse with satellite cells and form myotubes (Figure 3E,F). In summary, we successfully sorted mesoangioblasts and fibro/adipogenic progenitors from freshly isolated skeletal muscle by FACS and characterized them, finding no differences in the expression of markers and in their in vitro differentiation potencies.

To check for receptors that match the growth factor footprint from the dystrophic cardiac and skeletal muscle, we subjected murine mesoangioblasts and fibro/adipogenic progenitors to a qPCR screening (methods). Some receptors were expressed on both mesoangioblasts and fibro/adipogenic progenitors such as *Ccr1*, *Ccr5*, *Pdgfra* and *Pdgfrb* (Figure 4A). In addition, we also observed the expression of several important cell-surface receptors such as *Cdh5*, *Itga2* and *Itgb2* (Figure 4A). We did not proceed with these receptors while we were interested in a migration axis shared between dystrophic tissue and interstitial stem cells. Via flow cytometry we validated the localization of Ccr5, Pdgfra and Pdgfrb while Ccr1 was not present (Figure 4B,C). In sum, we screened for complementary receptors on mesoangioblasts and fibro/adipogenic progenitors at a gene and protein level and confirm the presence of PDGFRA and PDGFRB on both and CCR5 on mesoangioblasts.

Because both mesoangioblasts and fibro/adipogenic progenitors express PDGFRB and its ligand is upregulated in dystrophic skeletal as well as cardiac muscle, we wanted to check if these cells are able to migrate towards this growth factor. We performed an in vitro migration assay on mesoangioblasts as well as fibro/adipogenic progenitors and assessed their ability to migrate towards PDGF-BB (Figure 5A). Both were able to migrate towards PDGF-BB albeit with different efficiencies (Figure 5B). Additionally, we also confirm the ability of fibro/adipogenic progenitors to invade through Matrigel, although with a lower efficiency as without Matrigel (Figure 5C). 

Next, we wanted to assess if the migration of these cells required the presence of the PDGFRB receptor. For this reason, we isolated fibro/adipogenic progenitors and mesoangioblasts from *Pdgfrb*-loxP mice (methods). After the transient addition of Cre (using ASLV-Cre-virus like particles), we assessed the presence of the PDGFRB receptor via flow cytometry and observed a drastic decrease (Figure 6A). When we subjected these cells to an in vitro migration towards PDGF-BB, we observed a dramatic drop in migration ability, demonstrating the crucial role of PDGFRB in the migration of mesoangioblasts and fibro/adipogenic progenitors to PDGF-BB (Figure 6B). Altogether, we provide proof that mesoangioblasts and fibro/adipogenic progenitors are able to migrate and invade towards PDGF-BB and that the PDGFRB receptor is essential for the migration towards this ligand.

In order to evaluate if this migration axis is also important for cell therapy, we isolated cMABs from human cardiac muscle biopsies and skmMABs from human skeletal muscle biopsies. MABs were isolated by explant outgrowth and sorted for tissue non-specific alkaline phosphatase, liver/kidney/bone (ALPL) [22] (Figure 7A). The isolation efficiency of MABs per biological replicate was dependent on their outgrowth and percentages of ALPL+ populations varied between five to sixty percent. Further characterization of skmMABs showed that they were all CD31 and CD45 negative, while they were positive for CD44 and NG2 (Figure 7B). cMABs differentiated into cardiomyocytes, as shown by immunofluorescence staining for Cardiac Troponin I (CTnI), however, the pattern of Connexin43 (Cnx43) showed that they were still premature (Figure 7C). skmMABs spontaneously formed myotubes upon myogenic differentiation (Figure 7D). We also showed that *CCR1*, *CCR3, CXCR4, FKN* and *MET* are expressed in cMAB and skmMABs and confirmed the expression of *PDGFRB* (Figure 7E). Human skin fibroblasts were used as a control cell type since they express the majority of these markers. In sum, we isolated and characterized human mesoangioblasts from cardiac and skeletal muscle and provide proof that they express *PDGFRB*.

## 3. Discussion

Cell therapy is one of the approaches that was suggested to treat muscular dystrophies. While several cell types, including mesoangioblasts and fibro/adipogenic progenitors, have shown to be able to engraft into the muscle and fuse with existing muscle fibres, the efficiency is still very low. Although quite some research has been performed on increasing the migration of MABs by pre-treatment with different growth factors [3,23], the signals secreted by the tissue that attracts these cells have not been investigated. Therefore, we performed qPCR assays on skeletal muscle for chemokines and growth factors. We found that a considerable number of chemokines and growth factors were upregulated in the skeletal muscle of *Sgca*-null mice. *CC*-chemokines showed the highest increase; these chemokines have also been reported in mdx mice as the main factors responsible for the attraction of macrophages and lymphocytes [24]. We also found a rise in the expression levels of *Sdf1*, *Cx3cl1* and the *Pdgf*-family growth factors. Sdf1 is known for its prominent role in migration and homing and pre-treatment with Sdf1 can also improve homing of cardiac mesoangioblasts [23]. PDGF-BB has also been reported to be upregulated in muscular dystrophy patients, is secreted by regenerative myofibers and infiltrating macrophages and is related to the progression of the disease [25,26]. Many chemokines are expressed in DMD myofibres while being negative in healthy muscle, SDF1 is highly expressed in the vascular endothelium in muscular dystrophy [27,28].

Subsequently, we analysed the chemokine profile of the heart of different animal models with limb girdle muscular dystrophy. We included C57Bl6, *Sgca*-null, IC *Sgcb*-null and ID *Sgcb*-null mice, ranging from none to extremely severe cardiac complications, respectively. We detected plenty of differentially expressed chemokines and chemokine receptors between dystrophic and healthy mice, however also within different types of dystrophy. Of these genes, a fairly low number were shared between dystrophies, meaning that most of the chemokines that were up- or down-regulated were unique to the type of dystrophy or the stage of heart failure. The benefit of the varying grade of cardiomyopathy that arises in these types of limb girdle muscular dystrophy is that we could also compare differentially expressed chemokines between models. In this respect, we revealed specific chemokine footprints connected to particular classes of limb-girdle muscular dystrophy. The most interesting finding was that *Sgcb*-null mice, suffering from a higher grade cardiomyopathy, expressed more upregulated chemokines compared to *Sgca*-null mice with *Cx3cl1*. Cx3cl1 is secreted during chronic cardiomyopathy and has been shown to drastically worsen cardiac complications [29,30]. So far, it has never been linked to cardiac complications in muscular dystrophy. In addition, Cx3cl1 is also a potent chemoattractant [31] and could be used as a target for cell therapy. Further analysis of differentially expressed chemokines between ID and IC *Sgcb*-null mice revealed a radical downregulation of chemokines and chemokine receptors as revealed by the GO enrichment analysis. This was expected considering the lack of adaptive immune system of the ID *Sgcb*-null mice. Additionally, we performed hierarchical clustering on differentially expressed genes between all disease groups and found five gene groups, all relating to a down or upregulation in a type of limb-girdle muscular dystrophy, further strengthening the heterogeneity of chemokines in muscular dystrophy that could reflect the stage of disease progression and, as such, be used as potential biomarkers.

Next, we needed to search for matching chemokine receptors on cells that are able to migrate. Therefore, we isolated mesoangioblasts and fibro/adipogenic progenitors from fresh murine skeletal muscle by means of FACS and demonstrate that PDGFRA and PDGFRB are expressed by mesoangioblasts as well as fibro/adipogenic progenitors. While we show that both *Pdgfa* and *Pdgfb* are upregulated in the heart and in skeletal muscle, it is mainly PDGF-BB that is linked to activation of resident stem cells and their migration in the skeletal muscle and other tissues [15,32,33]. Therefore, we tested the in vitro migration capacity of mesoangioblasts and fibro/adipogenic progenitors to PDGF-BB. We conclude that both cell types migrate efficiently to PDGF-BB, even when they had to invade through a layer of Matrigel. Furthermore, we proved that the migration towards PDGF-BB is mainly through its binding with PDGFRB. In fact, when we knocked out *Pdgfrb* by means of transient addition of Cre to freshly isolated cells from *Pdgfrb*-flox mice, the migration of interstitial cells was impaired. 

Ultimately, we isolated mesoangioblasts from human cardiac and skeletal muscle biopsies as ALPL positive cell population and screened them for chemokine receptors. We show that they also express several chemokine receptors, including *CCR1*, *CCR3, CXCR4, FKN* and *MET,* and we confirmed that *PDGFRB* is present in MABs [11]. Further experiments are necessary to prove that the PDGFB/PDGFRB axis is crucial for human interstitial stem cell migration. Altogether, our results provide a chemokine footprint of limb girdle muscular dystrophies and show for the first time the expression of fractalkine in severely affected cardiac and skeletal muscles. Finally, our data point to the PDGFB/PDGFRB axis as a crucial signalling pathway for MAB and FAP migration and a potential target to improve stem cell-based therapeutic potential. In conclusion, we propose that pre-treatment with PDGF-B can activate cells and let them migrate more efficiently, which has recently been shown for satellite cells [34], and this can be potentially translated to the clinical setting. Nevertheless, we believe that activating the PDGF-pathway should be done carefully because it is involved in many cell types and is highly active in cancer tissue.

## 4. Materials and Methods

### 4.1. Bioethics

All mouse experiments were conducted in strict accordance with European law and were approved by the Animal Ethics Committee of KU Leuven (P150/2014, approved on 7 July 2014 and P169/2017 approved on 7 November 2017). The mice were bred and maintained in the laboratory animal centre of the KU Leuven. For skeletal muscle growth factor screening, C57Bl/6J and C57Bl/6J *Sgca*-null mice of five-month-olds were used. For cardiac muscle growth factor screening, C57Bl/6J, C57Bl/6J *Sgca*-null, C57Bl/6J *Sgcb*-null and C57Bl/6J *Sgcb*/*Rag2*/*γc*-null mice between four to seven months old were used. For isolation of interstitial stem cells, C57Bl/6J and C57Bl/6J *Pdgfrb*^fl/fl^ of one-month-olds were used. The work on human cardiac and skeletal muscle biopsies was approved by the ethical commission of UZ Leuven ethical committee (study protocol: s57322) and conforms to the guidelines of the 2000 Helsinki declaration. The anonymized samples were collected under signed informed consent. Human skin fibroblasts were sent by the Laboratorio Colture Cellulari (Siena Italy).

### 4.2. Isolation of Murine Skeletal and Cardiac Muscles

Prior to dissection, mice were euthanised with CO_2_ intoxication or cervical dislocation. Cardiac muscle was collected by surgical dissection and were washed in PBS to remove excess blood. Skeletal muscle tissue collection was carried out as followed: skin around hindlimbs was removed, all hindlimb muscles were isolated by surgical dissection and tissue samples were frozen at −80 °C for RNA isolation or were finely minced for digestion.

### 4.3. RNA Isolation and Quantitative PCR

Skeletal and cardiac muscle tissues were purified to RNA with the TRIzol Plus RNA Purification Kit (Invitrogen #12183555, Merelbeke, Belgium) by following the standard protocol. Live cells were collected in RLT lysis buffer (Qiagen #1053393, Antwerp, Belgium) and flash-frozen on dry ice. Cell lysates were homogenised with 21g syringes before RNA isolation using the RNeasy Plus Micro Kit (Qiagen #74034). Reverse transcription was performed using the Superscript III reverse transcriptase (Invitrogen #18080093), following the manufacturer’s recommendations. Expression levels of mRNA were assessed by real-time PCR using the Platinum SYBR Green qPCR SuperMix-UDG (Invitrogen #11733038) on the fast run mode of the Viia7 Real-Time PCR system (Thermo Scientific, Merelbeke, Belgium). Negative controls were included, omitting the cDNA from the reaction. Cardiac muscle tissue samples were subjected to an RT^2^ Profiler PCR Array Mouse for Cytokines & Chemokines (#330231 Qiagen) according to manufacturer’s recommendations. For skeletal muscle tissue samples and cell samples, primers were manually selected and designed (Appendix A). mRNA expression was normalised to *Rpl13a* (for all mouse and human experiments, if not otherwise specified). Gene expression assays in Figure 1, Figure 2, Figure 3C, Figure 4A and Figure 7E show mean values over *n* = 3–6, and for biological replicates, experiments were performed three times.

### 4.4. Annotation and Functional Enrichment Analyses

Enrichment analysis was performed using the webserver Enrichr with default parameters [35] and annotations provided by GO cellular biology [20,21]. For Appendix A, the top six enriched terms were considered. Full results are available in Appendix A.

### 4.5. Digestion of Murine Skeletal Muscle

Skeletal muscle digestion was carried out in accordance to the protocol published by Liu et al. [36]. In short, all hindlimb muscles were isolated by surgical dissection and finely minced. Minced muscles were digested with Collagenase II (Sigma-Aldrich #234155, Darmstadt, Germany 700–800 U/mL in PBS) supplemented with 10 percent horse serum (HS, Gibco #26050088), 25 mM CaCl_2_ and 100 U/mL penicillin-streptomycin (Gibco #15140122, Merelbeke, Belgium) for 90 min at 37 °C in a warm water bath with agitation (60–70 rpm). At 60 and 90 min, the tissue was homogenised with an 18- and 20-gauge needle, respectively. Next, a red blood cell lysis was performed by incubating the pelleted cells in the red blood cell lysis buffer (154 mM NH_4_Cl, 10 mM KHCO_3_, 0.1 mM EDTA) for 5 min followed by a PBS wash, filtration through a 40 µm cell strainer (Corning #352340, Darmstadt, Germany) and centrifugation for 5 min at 400 g at 4 °C.

### 4.6. FACS Isolation of Murine Cells

The single-cell suspension was diluted to 1 x 10^6^ cells/mL with FACS buffer (2 percent foetal bovine serum (FBS, Gibco #10082147), 10mM HEPES and 10mM NaN_3_ at a pH of 7.2) and was supplemented for 30 min at 4 °C and protected from light with the following antibodies: anti-mouse Ter-119-FITC, anti-mouse CD31-FITC, anti-mouse CD45-FITC (Invitrogen, #11-0311-81, #11-0451-81, #11-5921-81, respectively) for selecting the Lin‒ population; anti-mouse Ly6a-APC (Invitrogen #17-5981-82), anti-human Alpl-PE (R&D Systems #MAB1448) and anti-mouse Itga7-PE-Vio770 (Miltenyi Biotec #130-102-718, Leiden, Netherlands) to enrich the Lin- population for FAP and MAB, respectively. All antibodies were titrated (Appendix A). The cells were washed twice and stained with Calcium Blue (Invitrogen #C1429, 1 µL per mL) and Hoechst 33258 (Invitrogen #H3569, 1 µL per mL) to assess viability. Cells were sorted at a concentration of 4 × 10^6^ cells/mL. Sorting was performed on the BD FACSAria III (BD biosciences, Erembodegem, Belgium), which was equipped with five lasers, using a 100 μm nozzle at 18 PSI. Compensation measurements were performed for single cell stains with UltraComp eBeads compensation beads (Invitrogen #01-2222-41). Sorting gates were strictly defined with fluorescent minus one (FMO) controls. The following gating strategy was applied while sorting the cells: first, the cells were selected based on their size and granularity or complexity (side and forward scatter), then only live cells were selected by gating for cells positive for Calcein blue and negative for 7-AAD, and then any events that could represent more than one cell were eliminated. Next, the Lin−(CD31−CD45−TER119−)Itga7– population was selected, followed by further analysed populations that were negative or positive for a given marker: Lin−Itga7+Ly6a+ (FAP), Lin−Itga7–Ly6a+Alpl+ (MAB) and Lin–Ly6a–Alpl–. Data were collected with the FacsDIVA (BD biosciences) software. Biexponential analysis was performed using FlowJo (Treestar, Erembodegem, Belgium) software. The mouse sorting experiments were performed three to four times (Figure 3A,B).

### 4.7. FACS Isolation of Human Cells

Human cardiac and skeletal muscle biopsies were washed in PBS and cut into 1–2 mm^2^ chunks. Muscle fragments were incubated in 3 cm^2^ petri dishes in growth medium: high glucose DMEM medium (Gibco #61965026) supplemented with 20 percent FBS (Gibco #10082147), 10 percent HS (Gibco #26050088) and 1 percent penicillin–streptomycin (Gibco #15140122). After 1–4 weeks, cells sprout from muscle biopsies. Upon a confluence of 80 percent, cells were collected and stained with anti-human Alpl-PE (1 in 40, R&D Systems #MAB1448) for 30 min at 4 °C protected from light. Afterwards, the single-cell suspension was diluted to 4 × 10^6^ cells/mL with FACS buffer (2 percent foetal bovine serum (FBS, Gibco #10082147), 10mM HEPES and 10mM NaN_3_ at a pH of 7.2.) and were sorted. Sorting was performed on the BD FACSAria III (BD biosciences), which was equipped with five lasers, using a 100μm nozzle at 18 PSI. Sorting gates were strictly defined with unstained controls. The following gating strategy was applied while sorting the cells: first, the cells were selected based on their size and granularity or complexity (side and forward scatter), then, only cells that were positive for Alpl (cMAB for cardiac muscle samples and skmMAB for skeletal muscle samples) were selected. Data were collected with the FacsDIVA (BD biosciences) software. Biexponential analysis was performed using FlowJo (Treestar) software. The mouse sorting experiments were performed three times (Figure 7A).

### 4.8. Ex Vivo Murine Adipogenic and Myogenic Differentiation

The same number of cells was sorted and seeded into flat-bottom 24 well cell culture plates (Corning #3738) coated with Collagen (Sigma-Aldrich #C9791), cultured to confluence in high glucose DMEM medium (Gibco #61965026) supplemented with 20 percent FBS (Gibco #10082147), 10 percent HS (Gibco #26050088) and 1 percent penicillin–streptomycin (Gibco #15140122). For adipogenic differentiation cells, we used the StemPro Adipogenesis Differentiation kit (Gibco #A1007001) with medium changed every 3–4 days. Differentiation was carried out for 10–12 days after induction at which point the cells were stained for imaging or were collected for RNA isolation. For myogenic differentiation cells, MAB, FAP and satellite cells (transduced with GFP transgenic MLV viral vectors (3.325 × 10^6^ TU/mL) provided by the Leuven Viral Vector core) were mixed with control satellite cells at a 1:2 ratio and treated with high glucose DMEM medium supplemented with 2 percent HS (Gibco #26050088) and 1 percent penicillin-streptomycin (Gibco #15140122) with medium changed every 1–2 days. The culture was carried out for 3–4 days after induction whereupon the cells were stained for imaging. Cells were kept in hypoxic culture conditions (5 percent O_2_, 5 percent CO_2_ and 37 °C).

### 4.9. Immunofluorescence

For adipogenic differentiation, differentiated cells were fixed with four percent paraformaldehyde before staining with Hoechst (nuclei), Oil Red O (lipids) and anti-mouse Perilipin A/B with secondary antibody anti-rabbit AF488 (Figure 3E, Sigma-Aldrich #SAB4600234). Fluorescence images were acquired at 20× magnification via Eclipse Ti inverted microscope (Nikon, Amsterdam, Netherlands). We acquired multiple independent wells per experiment. One well is represented (Figure 3E). Image analysis was performed in ImageJ/Fiji, lipid droplets (red) and nuclei (blue) images were filtered using a Gaussian blur (sigma equal to 2) before an automatic thresholding. The automatic thresholding algorithm selections were chosen on the basis of visual inspection of output images. For myogenic differentiation, differentiated cells were fixed with four percent paraformaldehyde before staining with Hoechst 33342 (nuclei, Sigma-aldrich #14533) and MF20 (Myh1, hybridoma bank). Fluorescence images were acquired at 10× magnification via an Eclipse Ti inverted microscope (Nikon).

### 4.10. Flow Cytometry Analysis of Murine Cells

The same number of cells was seeded into flat-bottom T75 cell culture flasks (Sigma-Aldrich #CLS3290) coated with 0.1 percent Collagen (Sigma-Aldrich #C9791), cultured to confluence in high glucose DMEM medium (Gibco #61965026) supplemented with 20 percent FBS (Gibco #10082147), 10 percent HS (Gibco #26050088) and 1 percent penicillin–streptomycin (Gibco #15140122). The single-cell suspension was diluted to 1 × 10^6^ cells/mL with FACS buffer (2 percent foetal bovine serum (FBS, Gibco #10082147), 10mM HEPES and 10mM NaN_3_ at a pH of 7.2) and supplemented for 30 min at 4 °C protected from light with the following antibodies: anti-mouse CCR1-APC (R&D systems #FAB5986A), anti-mouse CCR5-PE (Invitrogen #12-1951-82), anti-mouse PDGFRA-APC (Invitrogen #17-1401-81) and anti-mouse PDGFRB-APC (Invitrogen #17-1402-82), anti-mouse CD34-eFluor450 (Invitrogen #48-0341-82), anti-mouse CD44 (Invitrogen #17-0441-82). All antibodies were titrated (Appendix A). Analysis was performed on a BD FACSCanto II HTS (BD biosciences) equipped with three lasers. Data were collected with the FacsDIVA (BD biosciences) software. Biexponential analysis was performed using FlowJo (Treestar) software. The mouse sorting experiments were performed once (Figure 3D,E, Figure 4B,C).

### 4.11. In Vitro Transwell Migration

Migration experiments were performed using 24 and 96 well receiver plates (Corning Costar, Darmstadt, Germany) in combination with 6.5 and 4.26 mm diameter inserts with 8 µm pores (Corning Costar). Inserts were coated with 0.1% Collagen for migration assays or with Matrigel (200–300 µg/mL in coating buffer (0.01 M Tris (pH = 8.0), 0.7% NaCl), Corning) for in vitro invasion assays. Receiver wells were filled with chemoattractant depending on plate design (assay buffer (50 mM HEPES (Gibco), 0.5% FBS (Gibco) in HBBS (Gibco)), 30% FBS in PBS (Gibco) or PDGF-BB (12.5 ng/mL or 25 ng/mL in assay buffer)). 2 × 10^5^ cells were seeded on 24 well plate inserts and 9 × 10^4^ cells were used for 96 well plate inserts. Migration plates were incubated in hypoxic conditions (5% O_2_, 5% CO_2_) at 37 °C for 8 h.

### 4.12. Quantification of In Vitro Transwell Migration

For visual inspection of migration, inserts were fixated for 10 min in 70% ethanol and stained in 0.2% Crystal Violet for 10 min. Fluorescence images were acquired at 20× magnification via an Eclipse Ti inverted microscope (Nikon). We acquired multiple independent wells per experiment. One well is represented (Figure 5A). For quantification, we used flow cytometry. Cells remaining on the upper side of the insert filter were removed using cotton swabs, the lower side of the insert was washed twice with PBS and cells were detached with 0.25% Trypsin-EDTA (Gibco #25200056). Then, 123 counts eBeads Counting Beads (Invitrogen #01-1234-41) were added (1:3 ratio) and samples were counted with the BD FACSCanto II AIG. The absolute number of migrated cells was calculated with the following formula:$$Absolute\;Count\;\left(\frac{cells}{µL}\right)=\frac{\left(Cell\;count*eBead\;Volume\right)}{\left(eBead\;count*Cell\;Volume\right)}\times eBead\;concentration$$

### 4.13. Viral Vector-Mediated Knockout of Pdgfrb

For creating a population of PDGFRB- cells, FAP plated 10,000 cells/cm2 in high-glucose DMEM supplemented with 20 percent FBS (Gibco #10082147), 10 percent HS (Gibco #26050088) and 1 percent penicillin-streptomycin (Gibco #15140122) together with ALSV vectors containing recombinant Cre protein (ALSV-Cre). Viral vectors were cloned and produced by the Leuven Viral Vector core. After 72 h, cells were collected for determining knock-out efficiency and transwell migration (v.s. “In vitro transwell migration”). Knock-out efficiency was performed by staining with PDGFRB-APC antibody (#17-1402-82, Invitrogen) and flow cytometry analysis (Figure 6A). The experiment was replicated three times.

### 4.14. Other Computational Analyses and Data Processing Remarks

All computational analyses were performed with R version 3.5 and Bioconductor version 3.7. All *t*-tests and Wilcoxon rank-sum tests were unpaired and two-sided, if not otherwise specified. Fold change differences were calculated by means of the Livak method [37]. Differential expression was done by performing an independent *t*-test between two groups. For normalized expression values of cardiac muscle tissue, a Z-score was calculated from differentially expressed genes and hierarchical clustering was done on genes and samples. Venn diagrams were generated with limma 3.36.5 [38]. Heatmaps were generated with Pheatmap 1.0.12. All box plots were generated and displayed in R, using the geom_boxplot() function of ggplot2 with default parameters. The median value is indicated with a black line and a coloured box (hinges) is drawn between the first and third quartiles (interquartile range, IQR). The whiskers correspond to approximately 1.5× interquartile range (±1.58 interquartile range divided by the square root of *n*) and outliers are drawn as individual points. Individual points or median value is portrayed by a point. All bar plots display mean values as centres and the standard deviation as error bars. All included microscopy images and macroscopic images are representative.

### 4.15. Code Availability

Sample scripts used to process the data are available at https://github.com/SCIL-leuven/PDGFRB_muscle.

## Figures and Tables

**Figure 1 ijms-20-01118-f001:**
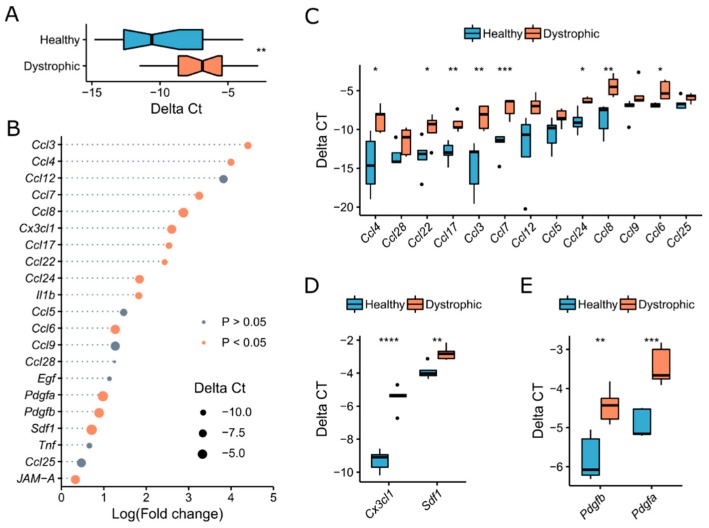
A general increase of chemokine expression in dystrophic skeletal muscle. (**A**) Boxplot of the average expression from 21 genes between the skeletal muscle of C57Bl6 (Healthy) and *Sgca*-null (Dystrophic) mice. *n* = 21, Wilcoxon rank-sum test. (**B**) Bubble chart displaying log fold increase of chemokine expression of Dystrophic compared to Healthy skeletal muscle. *n* = 5. (**C**–**E**): Expression of CC-chemokine family genes (**C**), CXC and CX3C family genes (**D**) and PDGF family genes (**E**) in dystrophic compared to healthy skeletal muscle from (**B**). Expression values as Δ*C*t, normalized to housekeeping genes (*Rpl13a*, *Rab35*, *Psma3*). *n* = 5. * *p* ≤ 0.05, ** *p* ≤ 0.01, *** *p* ≤ 0.005, **** *p* ≤ 0.001, *t*-test.

**Figure 2 ijms-20-01118-f002:**
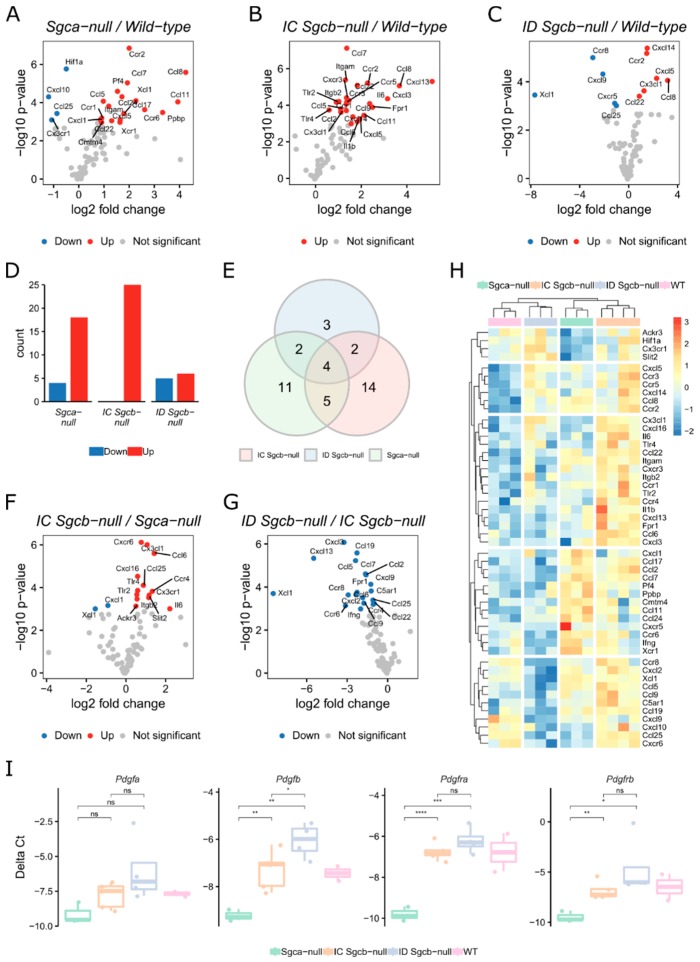
Comparison of chemokine expression in the hearts of different dystrophic animal models. Volcano plot of chemokine expression analysis on the hearts of Wild-type versus *Sgca*-null (**A**), IC (immunocompetent) *Sgcb*-null (**B**) and ID (immunodeficient) *Sgcb*-null (**C**). (**D**) Quantification of significant up- and down-regulated genes of (**A**–**C**). (**E**) Venn diagram comparing the significant differentially expressed genes of (**A**–**C**). (**F**) Volcano plot of chemokine expression analysis on the heart of IC *Sgcb*-null versus *Sgca*-null mice. (**G**) Volcano plot of chemokine expression analysis on the heart of ID *Sgcb*-null versus IC *Sgcb*-null mice. (**H**) Heatmap (blue-to-red) showing Z-score of significant genes from (**A**–**C**,**F**,**G**); hierarchically clustered rows and columns. (**I**) qPCR analysis of *Pdgf* genes; values shown as Delta Ct normalized to *Rpl13a*. *n* = 2–5. In (**A**–**C**,**F**,**G**), significant differentially genes (*p* ≤ 0.05) are coloured blue (downregulated) and red (upregulated). Genes are normalized to housekeeping genes (*Actb*, *Gusb*, *Gapdh* and *Hsp90ab1*). *n* = 3–4, *t*-test. * *p* ≤ 0.05, ** *p* ≤ 0.01, *** *p* ≤ 0.005, **** *p* ≤ 0.001, *t*-test.

**Figure 3 ijms-20-01118-f003:**
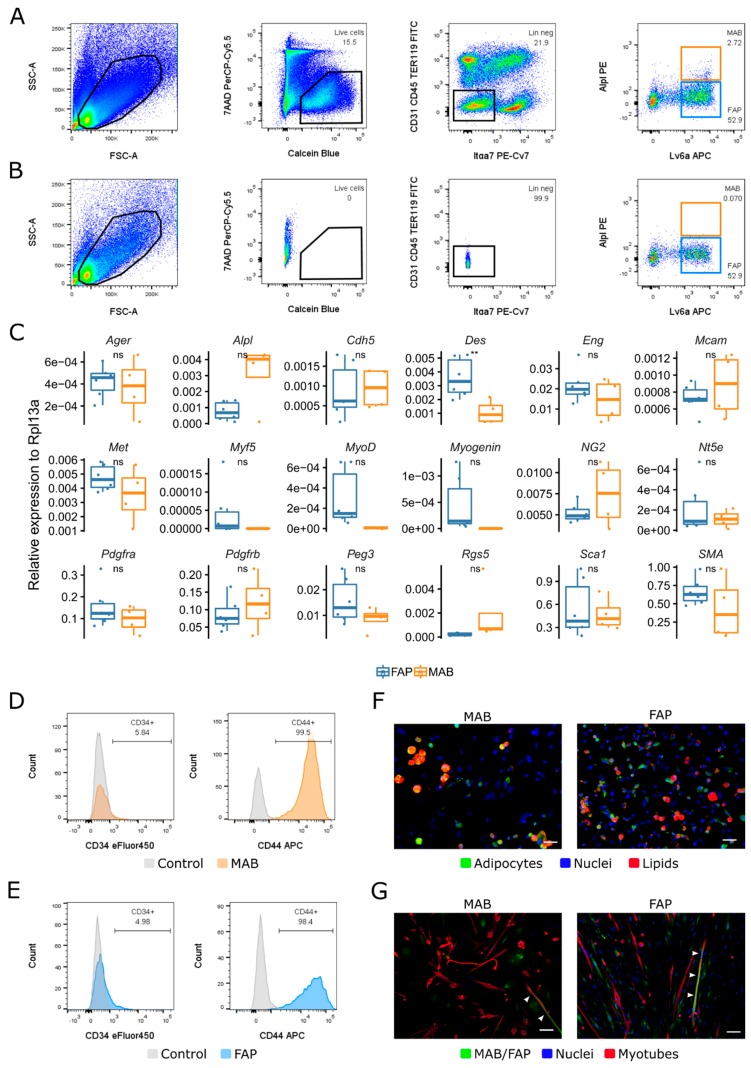
Isolation and characterization of interstitial stem cells from murine skeletal muscle. Gating strategy for FACS (fluorescent activated cell sorting) isolation of murine MABs (mesoangioblasts) and FAPs (fibro/adipogenic progenitors) (**A**) and control gates (**B**). (**C**) qPCR analysis of characteristic genes; values shown as relative expression to *Rpl13a*. *n* = 4–6. (**D**,**E**) Flow cytometry analysis of characteristic markers. MAB in orange (**D**), FAP in blue (**E**) and unstained control samples in grey. (**F**) Microscopy images of adipogenic differentiation; nuclei are stained with Hoechst (blue), lipids are stained with Oil Red O (red) and adipocytes are stained with Perilipin (green). (**G**) Microscopy images of myogenic differentiation from the co-culture of mouse MAB or FAP and satellite cells; Myotubes are stained with MyHC (red), MAB or FAP are stained with GFP (green) and nuclei are stained with Hoechst (blue). In (**E**,**F**), scale bar, 50 µm. * *p* ≤ 0.05, ** *p* ≤ 0.01, *t*-test.

**Figure 4 ijms-20-01118-f004:**
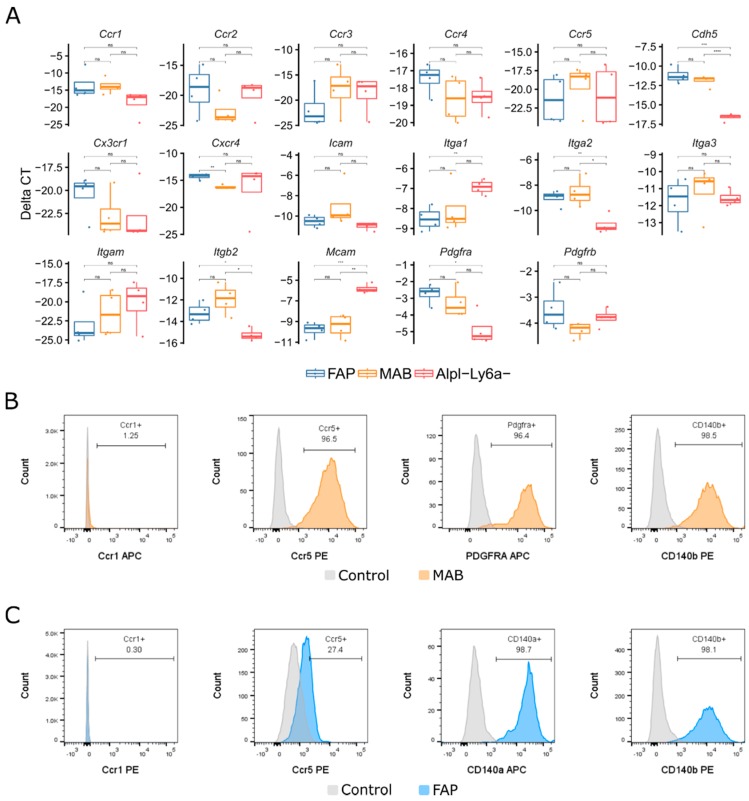
Chemokine receptor screening of interstitial stem cells. (**A**) qPCR analysis of chemokine receptor genes; values shown as Δ*C*t, normalized to *Rpl13a*. *n* = 4. (**B**,**C**) Flow cytometry for chemokine receptors localization on murine MAB (**B**) and FAP (**C**). MAB in orange, FAP in blue and unstained control samples in grey. * *p* ≤ 0.05, ** *p* ≤ 0.01, *** *p* ≤ 0.005, **** *p* ≤ 0.001, *t*-test.

**Figure 5 ijms-20-01118-f005:**
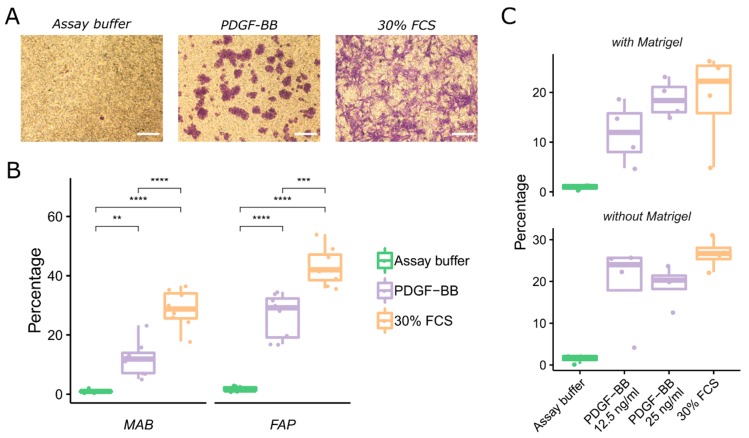
Interstitial stem cells migrate towards PDGF-BB. (**A**) Crystal violet staining of mesoangioblasts after transwell migration to the assay buffer (negative control), PDGF-BB and 30% FCS (positive control). Representative images shown. (**B**) Percentage of migrated cells shown in (**A**); *n* = 8. (**C**) Percentage of migrated fibro/adipogenic progenitors to different concentrations of PDGF-BB over Matrigel or no Matrigel-coated transwells; *n* = 4. * *p* ≤ 0.05, ** *p* ≤ 0.01, *** *p* ≤ 0.005, **** *p* ≤ 0.001, *t*-test. Scale bar, 50 µm.

**Figure 6 ijms-20-01118-f006:**
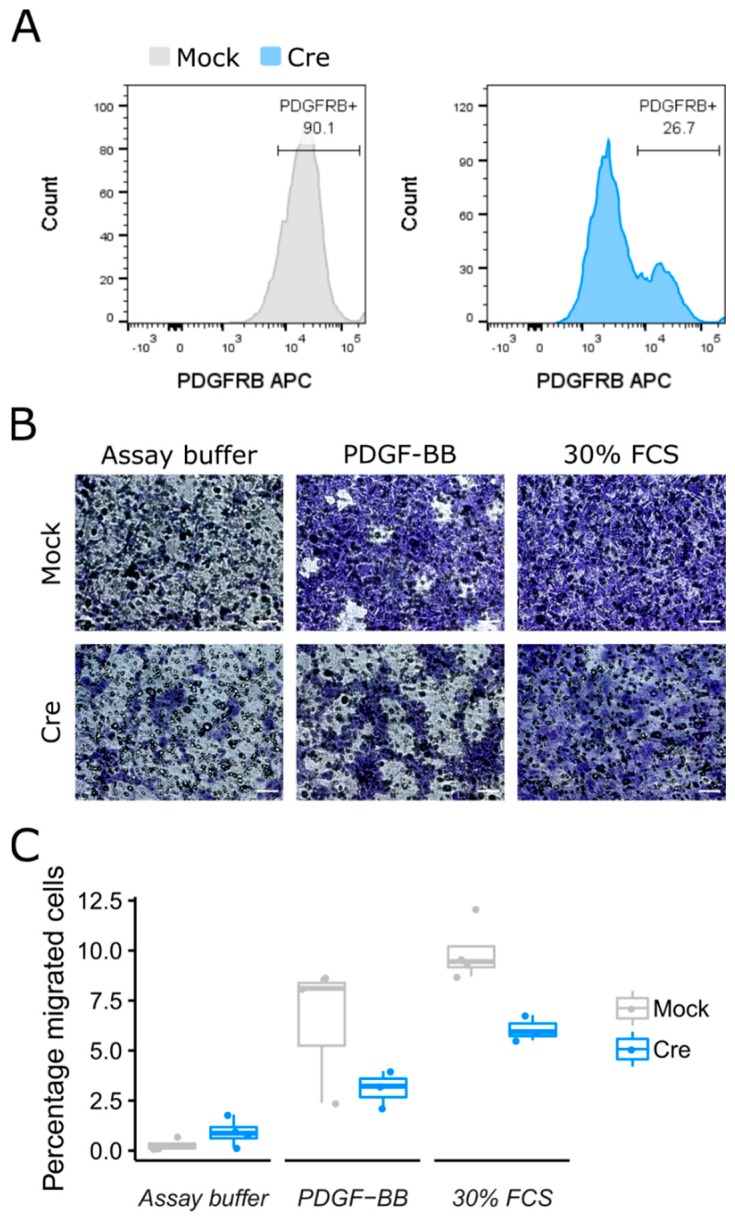
Interstitial stem cells migrate towards PDGF-BB through PDGFRB. (**A**) Flow cytometry analysis for PDGFRB of fibro/adipogenic progenitors isolated from the skeletal muscle of Pdgfrb-flox mice that were transiently transduced with ASLV-Cre or mock. (**B**) Crystal violet staining of transwell assays performed with cells from (**A**). (**C**) Percentage of migrated cells shown in (**B**). *n* = 3–4 independent wells. Experiments were repeated at least three times, yielding similar results. Scale bar, 50 µm.

**Figure 7 ijms-20-01118-f007:**
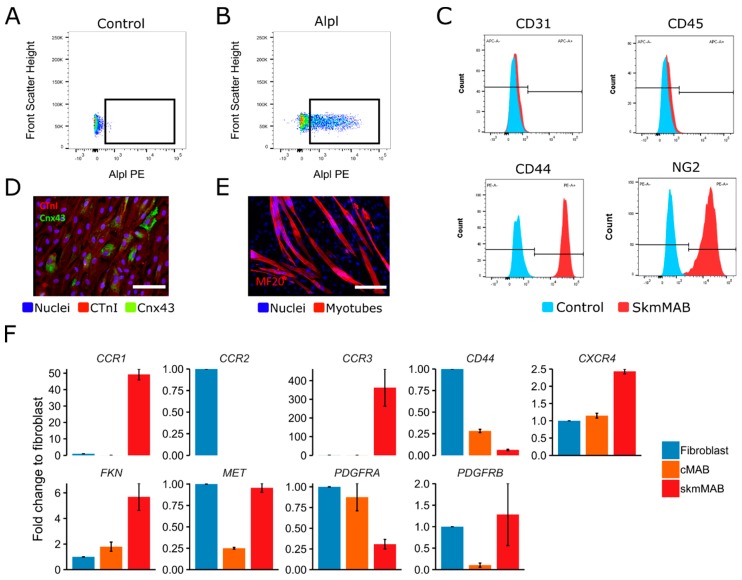
Isolation, characterization and chemokine receptor screening in human interstitial stem cells. (**A**,**B**) Gating strategy for FACS isolation of human MAB (**A**) and control (**B**). (**C**) Flow cytometry analysis of characteristic receptors. (**D**) Microscopy images of cardiac differentiation of cMAB; cardiomyocytes are stained with CTnI (red) and Cnx43 (green), nuclei are stained with Hoechst (blue). (**E**) Microscopy images of myogenic differentiation of skmMAB; myotubes are stained with MyHC (red), nuclei are stained with Hoechst (blue). (**F**) qPCR analysis of chemokine receptor genes; values shown as fold change expression ± S.E.M. compared to fibroblasts, normalized to *RPL13A*. *n* = 2–3. In (**D**,**E**), scale bar, 50 µm.

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
