# Peer review of "Growth Factor Screening in Dystrophic Muscles Reveals PDGFB/PDGFRB-Mediated Migration of Interstitial Stem Cells"

_ijms, 2019, doi:10.3390/ijms20051118_

Round 1
Reviewer 1 Report
The authors described the chemokine profile in different animal models of muscle diseases.
The results are clearly presented and the conclusions are supported by data.
Minor point
I suggest to indicate the rationale to performe the different experiments in five months-old mice. Is this age correlated to a specific disease's stage?
Moreover, the authors in another set of experiments used seven months-old mice. It is worth to justify the reason to use these animals age and to indicate any relation with the disease.
Author Response
Point 1: I suggest to indicate the rationale to perform the different experiments in five months-old mice. Is this age correlated to a specific disease's stage?
Response 1: For the chemokine screening of Sgcα-null mice, 5 months-old mice were chosen so that the dystrophic effects would be sufficient to affect the skeletal muscle environment. This is supported by the paper about the generation of Sgcα-null mice (Duclos et al JCB 1998), where they only describe signs of small foci of myocyte necrosis surrounded by atrophic and hypertrophic fibres, central nucleation, and fibre splitting from the age of 16 weeks. This rationale for the age selection of the Sgcα-null mice has been added to line 84-87.
Point 2: Moreover, the authors in another set of experiments used seven months-old mice. It is worth to justify the reason to use these animals age and to indicate any relation with the disease.
Response 2: For the chemokine screening of Sgcβ-null mice, 7 months-old mice were chosen so that the dystrophic effects in the heart would be sufficient to affect the cardiac muscle environment. This is supported by the paper where they generate and describe the pathology of the Sgc -null mice (Durbeej et al Mol Cell 2000), where in 30-week-old hearts, large areas of fibrosis were detected. This rationale for the age selection of Sgcβ-null mice has been added to line 115-116.
Reviewer 2 Report
In the present manuscript, Camps et al. used mouse models for muscular dystrophies at different stages to analyze the disease-dependent differential expression of growth factors/chemokines and their receptors in skeletal and heart muscle. Besides a series of differentially expressed mediators, it is basically found that PDGF-BB is a main factor inducing migration of mesoangioblasts and fibro-/adipogenic progenitors, which mainly respond via PDGFRB. These findings give rise to the hypothesis that PDGF may be used to develop a therapeutic strategy aiming at enhancing the effectivity of stem cell therapy in muscular dystrophies.
The manuscript is very well written and clearly describes the analyses performed and results obtained thereof. These results are fairly discussed in the context of the current literature and doubtless support the conclusion drawn. In summary, the manuscript is well suited to be published in IJMS, however, prior to that some minor corrections should be considered.
1. Some few typos and formal incorrectness should be revised.
2. Are there any data of the mediators’ expression on the protein level. These would strengthen the conclusion tremendously.
3. It may be discussed, which is the cellular source of the differentially expressed mediators, esp. of PDGF.
4. The discussion may conclude with a potential strategy how to affect the PDGF-PDGFR axis in a therapeutic fashion.
Author Response
We thank the reviewer for the detailed revision of the manuscript and we are certain it helped us to improve the quality. Please find the responses to your concerns below.
Point 1: Some few typos and formal incorrectness should be revised.
Response 1: We thank the reviewer for this observation and we corrected several typos throughout the manuscript: line 47, 57, 157, 210, 219, 227, 237, 291, 422
Point 2: Are there any data of the mediators’ expression on the protein level. These would strengthen the conclusion tremendously.
Response 2: Our intention for this study was to perform a screening for chemokines and growth factors in the skeletal and cardiac muscle of different muscular dystrophy animal models. We matched the mediators that we found from the screening with protein level data by flow cytometry on the interstitial stem cells. For the protein expression of PDGF-B in the dystrophic muscle we did not perform any immunostainings, however it is known that PDGF-B is highly expressed in dystrophic skeletal muscle (Zhao et al J Pathol 2003). We included this reference in the revised manuscript and highlighted this aspect.
Point 3: It may be discussed, which is the cellular source of the differentially expressed mediators, esp. of PDGF.
Response 3: We discussed more in detail the cellular source of the differentially expressed mediators in the revised manuscript. A part about expression of chemokines, including PDGF, in dystrophic muscles with two new references has been added to line 265-269.
Point4: The discussion may conclude with a potential strategy how to affect the PDGF-PDGFR axis in a therapeutic fashion.
Response 4: We thank the reviewer for this nice comment. Indeed, the perturbation of PDGF-PDGFR axis could be an intriguing therapeutic prospective to provide a more efficient migration of stem cells. Thus, we included a paragraph in the revised discussion considering also possible pitfalls (line 318-324).
Reviewer 3 Report
Comments:
This paper provides qPCR screen and some bioinformatic data regarding the chemokine pathways in different types of dystrophic muscle or cardiac muscle, which can be useful for the scientific society. And the interstitial stem cell migration response to PDGFRb pathway is convincing. However, the design of some experiments and the data interpretation can be improved. For example, I can’t see the significance of the qPCR comparison between the IC Sgcb and ID Sgcb mouse. It is expected to see differences in the immunologic pathway between these two types of mice. The interpretation of this part of result is not clear.
In addition, some experimental procedures and some results are not described clearly.
1. In line 420-421, it was written “ All antibodies were titrated (Supplementary table 1).” But can’t find the table in the supplementary doc.
2. Figure 3B is not clear. I can’t understand how the negative control gate was set, especially there seemed no live cells. And in the method section, line 360-362, the authors described that the “live cells were selected by gating for cells positive for Calcein blue and negative for Hoechst”, this is not right, as all the cells should stain for Hoechst, live and dead. Furthermore, the description: “and then any events that could represent more than one cell were eliminated” is confusing.
3. I can’t see how the authors concluded that “qPCR screen showed that: some receptors were expressed on both MAB and FAP such as Ccr1, Ccr5, Pdgfa and Pdgfb (Figure 4A) (line 178-180). As from figure 4A, there are other genes such as cdh5, Itga2, Itga3 expressed by both cell lines. What is your reference gene? Do you mean in comparison to alp-Ly6a- cell populations?
4. In terms of expression of markers by the MABs and FAPs, in line 165, the author claim that there were no differences between them, but Figure 4B data suggest that at least the Ccr5 expression is significant different.
5. Figure 7, first of all, the expression of PDGFRb on cMAB and skMAB have been well characterized and reported previously, this finding is not new. Secondly, the authors use qPCR method to identify the expression of chemokine receptors, but the way they present their data is confusing. Normally people use ddCT to compare the differential expression of a gene, using untreated samples or a certain type of cell population as control or baseline. But here they used housekeeping gene as control, I am not convinced by the way they claim as positive expression. There is a lack of explanation to interpret the data, either in the main text or in the method.
Author Response
We thank the reviewer for the detailed revision of the manuscript and we are certain it helped us to improve the quality. Please find the responses to your concerns below.
Point 1: I can’t see the significance of the qPCR comparison between the IC Sgcb and ID Sgcb mouse. It is expected to see differences in the immunologic pathway between these two types of mice. The interpretation of this part of result is not clear.
Response 1: The differential expression analysis between ID Sgcβ-null and IC Sgcβ-null mice was done with the idea in mind to look for chemokines that were downregulated. These chemokines can be linked to the immune system while we can clearly see that there are chemokines that are not downregulated but upregulated in IC and ID Sgcβ-null mice when compared to Wild-type mice. One example is Cx3cl1 or Fractalkine that is up-regulated in IC and ID Sgcβ-null skeletal muscles compared to Wild-type samples. Due to this comparison we can conclude that Cx3cl1 is solely expressed by the heart and not by the immune system. We now stress this aspect in the results to make this part clearer (line 127-133).
Point 2: In line 420-421, it was written “All antibodies were titrated (Supplementary table 1).” But can’t find the table in the supplementary doc.
Response 2: We apologise for this missing information. Revised Supplementary table 1 that includes antibody titrations is now attached as a supplementary document. Please find the table as attachment in the revised manuscript (on page 23).
Point 3: Figure 3B is not clear. I can’t understand how the negative control gate was set, especially there seemed no live cells. And in the method section, line 360-362, the authors described that the “live cells were selected by gating for cells positive for Calcein blue and negative for Hoechst”, this is not right, as all the cells should stain for Hoechst, live and dead. Furthermore, the description: “and then any events that could represent more than one cell were eliminated” is confusing.
Response 3: The name of the x-axis in figure 3B was accidentally wrong. It should be Calcein Blue instead of Hoechst. Calcein is a cell-permeable esterase substrate that can serve as a viability probe that measures both enzymatic activity as well as membrane integrity. In this view if the cells are still alive and functional an esterase reaction will cleave of the AM probe and Calcein blue becomes highly fluorescent, in this way we can gate “only” live cells. This has been also adapted in the methods section (line 376, line 391).
Point 4: I can’t see how the authors concluded that “qPCR screen showed that: some receptors were expressed on both MAB and FAP such as Ccr1, Ccr5, Pdgfa and Pdgfb (Figure 4A) (line 178-180). As from figure 4A, there are other genes such as cdh5, Itga2, Itga3 expressed by both cell lines. What is your reference gene? Do you mean in comparison to alp-Ly6a- cell populations?
Response 4: All genes in figure 4A are normalized to Rpl13a. We decide to go on with Ccr1, Ccr5, Pdgfra and Pdgfrb because they are receptors that are known for their capacity to migrate and because we found corresponding ligands in our expression assays of the skeletal and cardiac muscle. For example, CCL7 binds to CCR1, CCL5 to CCR5, PDGFA/PDGFB to PDGFRA/PDGFRB. We did not continue with genes such Cdh5, Itga2 and Itga3 because they are important in the cell-surface binding process of migration. We wanted to investigate a potential migration axis shared between tissue and cell types. However, because this information is imported to mention, we did include a remark on the expression of these cell surface mediators on the interstitial stem cells in the revised manuscript (line 193-196).
Point 5: In terms of expression of markers by the MABs and FAPs, in line 165, the author claim that there were no differences between them, but Figure 4B data suggest that at least the Ccr5 expression is significant different.
Response 5: We thank the reviewer for picking up this inconsistency in our manuscript and we agree that Ccr5 expression is significantly different in MAB compared to FAP. Thus, we adapted our paragraph summary of figure 4 in line 195, where we include that a difference in Ccr5 expression is observed between MAB and FAP.
Point 6: Figure 7, first of all, the expression of PDGFRb on cMAB and skMAB have been well characterized and reported previously, this finding is not new. Secondly, the authors use qPCR method to identify the expression of chemokine receptors, but the way they present their data is confusing. Normally people use ddCT to compare the differential expression of a gene, using untreated samples or a certain type of cell population as control or baseline. But here they used housekeeping gene as control, I am not convinced by the way they claim as positive expression. There is a lack of explanation to interpret the data, either in the main text or in the method.
Response 6: We agree that in general PDGFRb is a well-established marker for MABs. However, since we sorted MABs as Alpl positive cells we wanted to prove that this cell population still retains the pericyte markers together with other chemokine receptors that are not described yet. In this view it is a kind of quality control comparing to previous papers (i.e Dellavalle et al. Nat Cell Biol 2017), where MABs were obtained by limiting dilution cell cloning. We include this assumption in the revised results (line 246-249).
For the second point, we personally preferred dCT to represent qPCR data, since it can be possible to better understand the “absolute expression” of a certain gene (compared to a well -known housekeeping gene, in this case RPL13A). Negative controls are always included in our qPCR analysis, as now better indicated in our revised Methods (line 368-369). The fold induction representation, although I agree it is easy to interpret, can amplify differences due to the low amount of messenger RNA. Nevertheless, as suggested, we provided the revised figure 7 as ddCT (or fold change to the control cell type, in this case fibroblasts). However, the CCR3 expression has 500-fold induction in skmMABs compared to the baseline, although it is barely expressed in those cells (see delta CT in our original figure). This is the reason why we prefer dCT representation for qPCR data. Finally, we also modified the discussion (line 319-322, line 326-331) to provide a clear explanation to interpret the data.